# Ultrasonic Fatigue Testing of Structural Steel S275JR+AR with Insights into Corrosion, Mean Stress and Frequency Effects

**DOI:** 10.3390/ma16051799

**Published:** 2023-02-22

**Authors:** Yevgen Gorash, Tugrul Comlekci, Gary Styger, James Kelly, Frazer Brownlie, Lewis Milne

**Affiliations:** 1Weir Advanced Research Centre, Technology & Innovation Centre, University of Strathclyde, 99 George Street, Glasgow G1 1RD, UK; 2Weir Minerals South Africa, Weir Group, Isando, Johannesburg 1600, South Africa; 3Advanced Materials Research Laboratory, University of Strathclyde, Glasgow G1 1XQ, UK

**Keywords:** structural steel, very high cycle fatigue, ultrasonic fatigue, corrosion, frequency effect, mean stress correction

## Abstract

There are limited experimental data on VHCF for structural steels for >$10^{7}$ cycles. Unalloyed low-carbon steel S275JR+AR is a common structural material for the heavy machinery in minerals, sand and aggregate applications. The purpose of this research is to investigate the fatigue behaviour in the gigacycle domain (>$10^{9}$ cycles) for S275JR+AR grade steel. This is achieved using accelerated ultrasonic fatigue testing in as-manufactured, pre-corroded and non-zero mean stress conditions. As internal heat generation is a massive challenge for ultrasonic fatigue testing of structural steels which exhibit a pronounced frequency effect, effective temperature control is crucial for implementation of testing. The frequency effect is assessed by comparing the test data at 20 kHz and 15–20 Hz. Its contribution is significant, as there is no overlap between the stress ranges of interest. The obtained data are intended to be applied to the fatigue assessments of the equipment operating at the frequency for up to $10^{10}$ cycles over years of continuous service.

## 1. Introduction and Motivation

Unalloyed low-carbon structural steels (according to European standard EN 10025-2 [1]) are dominant materials to manufacture the components of heavy machinery used in the minerals, sand and aggregate industry. Despite good static strength, manufacturability and fatigue resistance, this group of steels has a pronounced strain rate effect. This is a significant challenge when producing SN curves using accelerated fatigue testing methods as key features, particularly the fatigue limit, strongly depend on the testing frequency. This research investigates the fatigue performance of S275JR+AR steel grade in the gigacycle domain ($10^{9}$–$10^{10}$ cycles) for applications that are intended to work for several years at an operating frequency of 15–16 Hz of loading with low-stress amplitude. However, accelerated fatigue testing (typically at 20 kHz) using ultrasonic machines significantly exaggerates fatigue strength compared to normal loading conditions. This issue needs to be addressed in the first instance of this research work.

The sand and aggregate and mining industry components are currently designed with high safety factors against SN curves with an assumed asymptotic fatigue limit above >$10^{7}$ load cycles. Nevertheless, fatigue cracks are seen even at a high number of cycles (>$10^{8}$), producing a large scatter in the data (over an order of magnitude) as the stress reduces. While high-cycle fatigue failure usually occurs at the surface, fatigue cracks at a very high number of cycles (>$10^{8}$) may initiate at oxides or intermetallic inclusions below the surface (or slag and flux inclusions in the case of welds). This transition in the failure mechanisms in the Very High-Cycle Fatigue (VHCF) regime has yet to be proved for the class of structural steels, including the S275JR+AR grade, which is the focus of this work.

Experimental investigation of the very high cycle domain with a number of cycles of >$10^{7}$ using conventional fatigue testing machines is costly. As such, it becomes practically impossible to reach the gigacycle domain, with the number of cycles of >$10^{9}$, without any acceleration of fatigue testing. One of the early comprehensive works [2], summarising the VHCF data and practical design approaches provided the material data and manufacturing-dependent recommendations. Furthermore, Sonsino [2] proposed some, material-specific SN functions for the VHCF domain that could predict fatigue performance in the gigacycle domain. Pyttel et al. [3] further discuss VHCF, giving an overview of the state of research, classifying metallic materials and influencing factors, and explaining different failure mechanisms in the VHCF region, like a subsurface failure, where the microstructural inhomogeneities play an essential role. Further recommendations for fatigue design of components are given by Pyttel et al. [3] and the status quo on describing the VHCF domain in various standards. The last monograph of Bathias [4] presented a thorough and authoritative examination of the coupling between plasticity, crack initiation and heat dissipation for lifetimes that exceed one billion cycles, leading to questioning the concept of the fatigue limit in VHCF, both theoretically and technologically. The review by Sharma et al. [5] provides insight into the recent developments in ultrasonic fatigue testing, its historical relevance, major testing principles and equipment. This work highlighted the significant defects, crack initiation sites, fatigue models and simulation studies to understand the crack development in the VHCF regime.

## 2. Available Fatigue Data

Available fatigue standards, e.g., BS 7608 [6], do not contain reliable experimental data up to $10^{9}$ cycles. Hence, the “fatigue design” of the critical machinery components is often carried out using stress amplitudes as low as 2% of the yield strength according to the design recommendations [7]. The available design SN curves [8,9] for structural steel grades [1] are limited to $10^{6}$ cycles and consider a fatigue limit at this threshold. Thus, currently, machines for sand aggregate and mining are most likely “over-designed” and hence not cost-effective.

The main problem is that there are limited data on VHCF for low-carbon structural steels, so it is difficult to make practical engineering predictions for the gigacycle domain ($10^{9}$–$10^{10}$ cycles). The existence of the plateau that characterises a transition from HCF to VHCF is an open question. An even more significant challenge is the interpretation and utilisation. The obtained ultrasonic data are subject to a considerable frequency sensitivity. This can be seen comparing the experimental SN diagrams with fatigue tests conducted at 110 Hz and 20 kHz for structural steels including C15E, C45E, and C60E [10] and fatigue tests at 10 Hz and 20 kHz for structural steel JIS S38C [11]. The fatigue limits identified at 20 kHz at around $10^{8}$ cycles to failure for S355J0 and S355J2 subgrades [12,13] are significantly higher than the recommended design values reported by the material databases [8,9] for $10^{6}$ cycles at low frequencies. On the other hand, corrosion either massively reduces or eliminates the fatigue limits for structural steels, as demonstrated in the example of R5 hot rolled low alloy steel grade [14], by the strong coupling between environment and cyclic loading even at an ultrasonic frequency.

The assumption that is applied to most metals, including structural steels, is that the fatigue lifetime before failure can be divided into the stages of crack initiation, crack propagation and final fast fracture [15]. In the range of European structural steels (S235, S275, S355, S460 and S500) [1], grade S275, with its relatively low yield strength and relatively high elongation at break appears more ductile rather than higher in strength. The correct assumption is that higher yield strength improves crack initiation fatigue resistance, which is opposite to higher ductility, which improves crack propagation fatigue resistance. For S275 steel loaded into the gigacycle regime, the question of which of these stages dominate the total fatigue life remains. This can be addressed by comparing the fatigue life of samples at the same stress amplitude with ideal surface finish to samples in the pre-corroded condition, where micro-cracks are believed to exist before the start of the test.

## 3. Experimental Procedure

### 3.1. Ultrasonic Machine

The number of cycles beyond $10^{7}$ can be attained in a viable period using the recently developed high-frequency testing machines, which typically run at 20 kHz. Evaluating the VHCF behaviour of materials is becoming increasingly important for applications such as mineral separation and transportation. As equipment in these sectors is required to last for years or even decades of continuous service, the number of load cycles they must withstand without failure will easily reach into the gigacycle domain (>$10^{9}$ cycles). To assess fatigue for $10^{9}$ cycles requires 1.6 years of normal testing at 20 Hz, which is not feasible. In contrast to that, when doing ultrasonic testing at 20 kHz, it would take only 0.6 days if intensive cooling is not required. Therefore, the central piece of the experimental setup is an ultrasonic fatigue testing system with a mean stress loading mechanism consisting of a standard Shimadzu USF-2000A machine and Shimadzu AG-X series (AG-X5kN) table-top autograph by Japanese company Shimadzu Corp. [16], with a maximum of 5 kN tensile load. Mean stress loading was applied to the 20 kHz samples using the AG-X5kN, which exerts a constant mean stress in the test sample by pulling it from both ends with the recommended force of ≤1.5 kN. Figure 1a shows the USF-2000A machine attached to the moving crosshead on one side and the frame base on the other side with a test sample in the middle. The standard air-cooling nozzles are pointed at the sample to suppress its intensive heating [17] caused by internal heat dissipation at around the resonance frequency of 20 kHz. The ultrasonic fatigue testing method implemented in Shimadzu USF-2000A machine was standardised by the Japan Welding Engineering Society [18,19]. This standard provides requirements for apparatus, specimens, test procedures, stress calculation methods and other necessary items. The data analysis carried out in the process of writing the standard [18,19] suggests that the ultrasonic fatigue testing is well suited to high-strength materials but not low-strength materials, which highlights the significant challenge in ultrasonic fatigue testing of structural steels, including grade S275.

### 3.2. Tensile Testing

Ultrasonic fatigue testing is based on the resonance loading by inducing longitudinal elastic waves in the specimen, with a peak in its central gauge location. Therefore, proper setup of the ultrasonic test requires accurate elastic properties of the tested material, as these properties directly define the stress amplitude and mean stress values applied to the sample. These properties were evaluated by carrying out tensile tests using an Instron 8802 servo-hydraulic fatigue testing system with an actuator force capacity of up to ±250 kN. Round tensile specimens 150 mm long were cut from the hot-rolled plate with a thickness of 12 mm, as shown in Figure 2a. The grip length sides are both 50 mm long, and the gauge length is 50 mm with fillets of radius R = 12.5 mm. The gauge diameter of the tensile specimens is $D_{g}$ = 6 mm. Strain was recorded using a strain-gauge attached using rubber o-rings as shown in Figure 2b. Three specimens were tested until failure with a loading rate of 1 mm/minute. The obtained experimental values for S275JR+AR grade were averaged and reported in Table 1 and compared to the values from the material quality certificate provided by the manufacturer and values from standards EN 10025-2 [1] and EN 1993-1-1 [20]. The comparison indicates insignificant deviation for all properties with Young’s modulus very close to the standard value from EN 1993-1-1 [20], a slightly better value of yield strength in the certificate compared to the experiment, and tensile strength and elongation at break values being very similar for the experiment and the certificate. However, both yield/tensile strength and elongation at break of the actual material are superior to those prescribed by the standard EN 10025-2 [1]. It should be noted that all the samples for all types of testing were cut from a single hot-rolled plate sized 1 × 2 m produced at Ilyich Iron and Steel Works of Mariupol, Ukraine – cast no. 293262-2, batch no. 20540.

### 3.3. Heat Generation

Heating is a massive challenge for ultrasonic fatigue testing [17], especially in the case of structural steels attributed with a pronounced frequency effect, including the S275JR+AR grade. As such, a temperature control arrangement (see Figure 1) is crucial for the proper implementation for testing. The use of intermittent driving with load blocks and cooling pauses was essential to address the intensive heating. The duration of the full-amplitude load block was 0.11 s for all implemented tests, but the duration of the cooling block was varied depending on the stress amplitude level. Additional inclusion of the VORTEC cold air gun, shown in Figure 1a, into the cooling arrangement helped to improve the thermal efficiency by producing significantly colder air. The temperature monitoring is done using a PyroCube thermometer from CALEX Electronics, which includes an infrared temperature sensor (PCU-S1.6-2M-1V), shown in Figure 1b, and configurable touch screen display for PyroCube (PM030), shown in Figure 1c. A sampling interval of 1 s, the maximum rate that can be achieved with the PM030, was used for temperature data logging at the beginning of the experimental program. The original experimental setup’s typical cooling air temperature was around 15 °C. The duration of the cooling pause was selected manually for each test depending on the measured temperature on the gauge surface of the sample using an infrared PyroCube thermometer. The temperature was controlled by watching the touch-screen display with the condition to keep it below 30 °C for all tests. This resulted in the maximum permitted variation of temperature of 15–30 °C in the loading block. Depending on the stress level, the cooling pause was varied from 0.5 s to 5 s to cool the sample surface down to 15 °C in the original temperature control arrangement. Extended pauses of up to 5 s were needed for high-stress amplitudes around 400 MPa. The combination of the loading block of 0.11 s and the pause of 5 s gives an equivalent frequency of 430.5 Hz as the worst-case scenario for the most intensive cooling demand. By adding the VORTEC cold air gun, it was possible to reduce the cooling pause to 0.1 s. However, the cooling became less stable with unpredictable temperature fluctuation and more sensitivity to the air supply. There was also a significant ice formation on the cooling hose for the long-duration tests. To decrease the sampling interval below 1 s, e.g., to 10 ms, and to initiate an emergency test stopping when temperature increases over 30 °C, the LabView program “Temp recorder and test stopper” has been developed, as shown in Figure 1d. The LabView program reads the temperature from the PyroCube thermometer, which is connected to the PC via an NI USB Multifunction DAQ data acquisition card.

### 3.4. Specimens Manufacturing

The ultrasonic sample geometry was manufactured following the standard WES 1112 [18] with a minimum recommended diameter of 3 mm in the gauge location. This diameter is half of the gauge diameter ($D_{g}$ = 6 mm) of the samples used in tensile testing and conventional fatigue testing. This difference in minimum diameter (3 mm vs. 6 mm) is considered to bring a limited size effect within 10% of the fatigue limit according to [21] for structural steels under VHCF conditions, with more conservative results for larger diameter. The shape of the ultrasonic sample is designed to resonate at 20 kHz and provide efficient air cooling within the allowable range of horn end displacements. Figure 3a shows the sample during the testing with the cooling nozzles and infrared sensor pointing at the middle of the specimen gauge section. Dimensions of the sample (shown in Figure 3b) were estimated using elastic properties of structural steel grades (S275 and S355), which are reported in Table 1. The surface finish of $R_{a}=0.8$ µm is considered a high grade close to “mirror finish” that requires very close control and higher costs. It is required for parts exposed to stress concentrations in industrial applications. To improve the accuracy of non-contact temperature monitoring, the samples have been painted in black matt colour using Rust-Oleum Stove & BBQ spray paint, as shown in Figure 3c. It has been practically identified that this coating provides a reliable adhesion to the metal surface and resistance to elevated temperatures. As manufactured, the surface of the gauge location has a high-grade finish of $R_{a}=0.8$ µm with an emissivity close to 0. The applied coating massively improves the emissivity, bringing it close to 1 and making the infrared temperature monitoring efficient. Finally, the dimensions of the ultrasonic samples are shown in Figure 3c.

### 3.5. Corrosion Inducement

To study the effect of corrosion on the fatigue resistance of S275JR+AR grade, two batches of pre-corroded samples have been prepared. They have the exact same dimensions (see Figure 3c), but they were exposed to 3.5% NaCl solution as the corrosion medium in 0.5 L beakers, as shown in Figure 4a. Figure 4b–d show the effect of corrosion on the surface of the sample after two weeks of “still seawater” treatment. The threads on the ends of the samples were protected from corrosion using RS PRO White PTFE thread seal tape that was 12 mm wide. Threads and adjacent areas were wrapped up in multiple layers of tape with different degrees of orientation, as shown in Figure 4b,c. This sort of waterproof isolation appeared to be quite reliable, as after removing the tape, the surface under it showed minor signs of corrosion, as can be seen in Figure 4d. When taken out of the water, samples have a thick rust layer, as shown in Figure 4b, but this layer is not mechanically stable and can be easily washed and wiped off. Removing the outer layer of greasy rust reveals a matt grey surface with an emissivity of >0.3, which is as expected for a heavily oxidised metal surface and is still suitable for infrared temperature monitoring. The pre-corroded surface is evenly covered with pits, resulting from material loss, which can be seen without additional magnification, as shown in Figure 4c,d. The surface roughness of the first batch, which was pre-corroded for two weeks, was measured using the surface roughness machine Mitutoyo SV 600 and appeared to be Ra=12.5 µm on average with a variation of ±0.5 µm. One benefit provided by testing pre-corroded samples over painted samples are that any cracks in the gauge section are visible during testing (see Figure 4e).

## 4. Results and Discussion

### 4.1. Results Summary

Fatigue testing was done at a conventional frequency of 15 Hz (Group 0) and ultrasonic frequency of 20 kHz to accelerate the testing process. Ultrasonic testing results can fall into four groups:Data points with a crack originating on the surface;Data points with a crack starting from subsurface (s/s);Data points for the samples pre-corroded for 2 weeks (p/c 2w);Data points for the samples pre-corroded for 1 month (p/c 1m).

The summary of the obtained fatigue testing results for S275JR+AR grade is shown in Figure 5 in the form of data points and corresponding SN curves. The SN curves are fitted to the experimental data using least squares regression and the classical Basquin’s equation [22] that describes high-cycle and very high-cycle low-strain behaviour:(1)$$\sigma_{a}=\frac{\Delta \varepsilon_{e}}{2}\;E=\sigma_{f}^{\prime }\;{\left(2N\right)}^{b},$$
where $\sigma_{a}$ is the amplitude of alternating stress; *E* is Young’s modulus; *N* is the number of cycles to failure and 2N is the number of reversals to failure; *b* is a fatigue strength exponent, for common metals −0.12<b<−0.05; $\sigma_{f}^{\prime }$ is a fatigue strength coefficient, which is approximately equal to the monotonic true fracture stress $\sigma_{f}$. Corresponding values of the fatigue material parameters (*b* and $\sigma_{f}^{\prime }$) and the values of coefficients of determination (R-squared) are reported in Table 2.

The contribution of the strain-rate effect on fatigue resistance is found to be very significant as there is no overlap between the stress ranges of interest. Low-frequency testing is done in the range of 200–275 MPa, while ultrasonic testing is in the range of 340–400 MPa for the condition of perfect surface finish. Data points at 15 Hz were obtained with the Instron 8802 servo-hydraulic fatigue testing system, which uses identical specimens for tensile testing but with a better surface finish. The obtained SN curve using least squares regression shows little scatter with R${}^{2}$ = 0.92 and looks entirely consistent compared to the available SN curves from material databases [8,9]. The low-frequency SN curve for S275JR+AR grade looks better than the lower bound of averaged fatigue data for JR, J0 and J2 subgrades of S275 from the Granta database [9], but worse than the 50% probability SN curve averaged for all subgrades from the FKM database [8].

Both groups of data points at 20 kHz with perfect surface finish demonstrate a relatively small scatter (R${}^{2}$ = 0.93 and R${}^{2}$ = 0.98 for groups 1 and 2, respectively, from Table 2) when fitted with least squares regression. They are shown in Figure 5. A major technical challenge comes from the intensive heat generation, especially when running tests at high-stress levels of 370–400 MPa. Figure 6 shows the temperature history of the sample tested at 400 MPa that accumulated over 4 million cycles before failure. The total testing life of the sample lasted over 8000 s or $2\frac{1}{4}$ h. It should be noted that the loading block lasted only 0.11 s compared to the cooling block varying from 2 to 5 s; more than 95% of the testing time was spent on cooling. It was possible to keep the temperature within the “room temperature” range of 15–30 °C for about half of the testing time while using the maximum cooling pause of 5 seconds. However, in the second part of the specimen life, an exponential temperature growth is seen with temperatures up to 200 °C in the loading block just before failure, even with the maximum cooling pause. During the accelerated crack growth stage with increased heat generation, it is not possible to control the temperature as the minimum duration of the loading block is 110 ms enforced by Shimadzu software [16]. When approaching the stress levels close to the fatigue limit (340 MPa), the temperature generation was significantly lower; therefore, the cooling pause could be reduced to 0.5 s after passing 1 billion cycles.

### 4.2. Fatigue Limit

The ultrasonic fatigue limit for S275JR+AR grade with a higher “mirror surface finish” of $R_{a}=0.8$ µm is found to be 340 MPa, as confirmed by the specimen that ran out after 10 billion cycles. The stress amplitude value of $\sigma_{lim}=340$ MPa was reached at 8.21 × 10${}^{7}$ cycles when extrapolating the SN curve with a crack originating on the surface or at 9.39 × 10${}^{8}$ cycles when extrapolating the SN curve with a crack starting from very close to the surface or subsurface. It can therefore be assumed that fatigue failure is unlikely when the stress is below 340 MPa and over 1 billion cycles have been accumulated. In the existing experimental batch, five samples were found as outliers, as they demonstrated an order of magnitude longer fatigue life compared to the primary Group 1 (see Figure 5). Obviously, the crack initiation process has been significantly delayed in those samples, and for this reason, they have been grouped into a secondary Group 2 and used to generate an assumed sub-surface SN curve. The assumption of sub-surface crack origin has been additionally investigated using a Yenway optical stereo-microscope, as shown in Figure 7, and a Hitachi scanning electron the microscope, shown in Figure 8. Colouring patterns in the form of circular marks on the optical microimages in Figure 7 indicate a crack initiation origin located under the surface. On the SEM images in Figure 8, this assumption is less evident as the fatigue progress colouring is not available. However, the radial marks on the fracture surface still point to the location which can be considered as subsurface or very close to the surface, which is similar to the findings [23,24] on fractographic features at the fatigue crack initiation sites.

The conventional fatigue limit at low frequency for S275JR+AR is expected to be around 210 MPa, which is higher than the values of 179 MPa from [9] and 195 MPa from [8]. It should be also noted that this fatigue limit of 210 MPa is confirmed at 10 million cycles, which is quite different to the fatigue limits from the material databases [8,9], which are only confirmed at 1 million cycles. The quantitative difference between 15 Hz and 20 kHz SN curves was measured in terms of stress amplitude as 167.7 MPa on average, which varied from 178 MPa at 1.0 × 10${}^{4}$ cycles to 154.6 MPa at 154.6 at 1.0 × 10${}^{10}$ cycles. This difference is used as a basic approach for frequency effect correction. Scaling down the fatigue limit from 340 MPa at 20 kHz using 167.7 MPa correction gives 172.3 MPa as a conservative prediction of the conventional fatigue limit at 15 Hz for S275JR+AR, which is close to values from the literature [8,9].

Figure 9a is assumed to show the evidence of continuous dynamic recrystallization, which is governed by the evolution of the dislocation structure under the cyclic load. This microstructure evolution process usually creates new grain boundaries from dislocation walls. This is quite evident in the right bottom image in Figure 9b that demonstrates grain fragmentation or refinement. Over the 10 billion cycles of stress induction at $\sigma_{lim}$ = 340 MPa, there was an evident reduction of frequency heat generation rate, which was quite rapid in the beginning of the test and saturated to a constant value around 1 billion cycles. In order to better understand the nature of the microstructural evolution caused by very high cycle fatigue, a deeper micro-structural analysis, including electron backscatter diffraction (EBSD), would be required.

### 4.3. Strain-Rate Effect

It is commonly understood that there is a significant strain-rate sensitivity present in BCC materials, which is caused by the resistance to dislocation motion at higher strain rates [25]. As such, materials which contain large BCC regions, such as the ferrite phases present in carbon steels, tend to exhibit significant frequency effects caused by this strain rate sensitivity [10,11,26,27]. It been shown by Bach et al. [10] that for a range of carbon steels, the strain rate sensitivity has a direct correlation with the volume content of ferrite. For steels with a greater ferrite content, a greater strain rate sensitivity was observed, which leads to a greater discrepancy between the SN curves produced at ultrasonic and at traditional frequencies. In order to evaluate the volume content of ferrite in the tested steels, image analysis was used on micrographs taken of the tested materials, following the procedures laid out in ASTM E1245-03 [28]. Sections were taken at a magnification of 200× from four random points in each material and the results were averaged to mitigate the influence of material variability: see Figure 9c. The ferrite content of the materials was then compared to the change in the fatigue limit at 20Hz and 20kHz in order to evaluate the effect of the ferrite content on the strain rate sensitivity. For the S275JR+AR subgrade, ferrite content evaluation based on four samples (A—80.9% B—81.1% C—84.1% D—83.1%) produced an average value of 82%, which is higher compared to other structural grades. It can therefore be seen that the material containing the higher ferrite content (S275JR+AR) had a more significant discrepancy between the SN curves at the two test frequencies; therefore, it had a higher strain-rate sensitivity. This agrees with the observations reported by Bach et al. [10], who reported that a greater ferrite content correlates to a greater strain rate sensitivity in the material. More discussion on the effect of the ferrite content on the frequency sensitivity and the heat generation is provided in [29].

### 4.4. Corrosion Effect

The pre-corroded samples demonstrated a significantly lower fatigue resistance with finite fatigue life of about 1.23 million cycles at the stress level corresponding to the fatigue limit $\sigma_{lim}$ = 340 MPa for the perfect surface finish. As seen in Figure 5, the fatigue performance dropped by 1.5 orders of magnitude in terms of cycles to failure; by over 50 MPa in terms of stress amplitude for 2 weeks of pre-corrosion; and by over 100 MPa in terms of stress amplitude for 1 month of pre-corrosion. This can be explained by the absence of the crack initiation phase in pre-corroded samples, as the crack grows directly from pits, as stress concentrators. Extrapolation of the obtained corrosion-fatigue SN curves is based on previous experimental work of the steel S355J0 [30] with a particular focus on corrosion and mean stress effects. Applying the assumption of the continuous damaging effect of the corrosion, power-law extrapolation predicts the design fatigue limits for 10 billion cycles as 242.1 MPa for 2 weeks of pre-corrosion and 177.4 MPa for 1 month of pre-corrosion, as shown in Figure 10. The frequency affect correction method discussed in Section 4.3 above can be applied to the extrapolated SN curves using the variable difference between ultrasonic and low-frequency SN curves. This scales down the SN curves to the low frequency and results in a conservative prediction of the design fatigue limit for 10 billion cycles of 87.5 MPa for 2 weeks of pre-corrosion and 22.8 MPa for 1 month of pre-corrosion, as shown in Figure 10.

### 4.5. Mean Stress Effect

The damage caused by a stress cycle depends not only on the stress amplitude, but also on the mean stress. When the tensile (positive) component of mean stress dominates, the stress cycle produces more fatigue damage, even if the stress amplitude is the same. The typical stress cycles in the heavy machinery for sand and aggregate and mining applications are generally symmetric. However, very often, gravity’s contribution can be significant and introduces a considerable asymmetry. The effect of mean stress on fatigue life is a long established research topic.

Historically, several methods have been proposed to describe its effects on the fatigue limit and, more generally, on material fatigue strength [31]. Comparisons of these methods have also been made to understand which better predict the fatigue behaviour in application to different structural materials [32]. A recent investigation [33] compared 11 classical mean stress correction methods and found that the Walker and the generalized Linear Formula perform best. Therefore, this research considers two approaches—Walker’s [32] and FKM [8], which is a variation of the linear formula. The equivalent stress, according to Walker’s method, can be assessed using these three equivalent formulas [32]:(2)$$\sigma_{ac}=\sigma_{a}{\left(\frac{2}{1-R}\right)}^{1-\gamma }\;\Leftrightarrow \;\sigma_{ac}=\sigma_{\max}{\left(\frac{1-R}{2}\right)}^{\gamma }\;\Leftrightarrow \;\sigma_{ac}={\sigma_{\max}}^{1-\gamma }\;{\sigma_{a}}^{\gamma },$$
where γ is the materials parameter for Walker’s mean stress correction.

The equivalent stress according to the FKM method [8] has the following form:(3)$$\sigma_{ac}=\sigma_{a}+M\cdot \sigma_{m},$$
where *M* is a correction factor, which defines the sensitivity to mean stress. The FKM approach can be presented in a form similar to conventional methods such as Gerber, Goodman, Soderberg and Morrow methods, which are based on ultimate strength $\sigma_{u}$, yield strength $\sigma_{y}$ or true stress at fracture $\sigma_{f}$ as limiting values of the mean stress:(4)$$\frac{\sigma_{a}}{\sigma_{ac}}+\frac{\sigma_{m}}{\left(\sigma_{ac}/M\right)}=1,$$
where $\left(\sigma_{ac}/M\right)$ is replaced with $\sigma_{u}$, $\sigma_{y}$ or $\sigma_{f}$ in classical mean stress correction methods.

The FKM method is based on the idea of using a line with the slope *M* in coordinates of mean stress $\sigma_{m}$ and stress amplitude $\sigma_{a}$ (or stress range $\Delta \sigma =2\;\sigma_{a}$). The slope coefficient *M* is generally related to fundamental material properties like Young’s modulus, yield strength and UTS for each class of steels or metal alloys. Application of the FKM mean stress correction method (Equation 3) to the Basquin Equation (Equation 1) results in the following equation for the equivalent stress amplitude:(5)$$\sigma_{ac}=\sigma_{f}^{\prime }\;{\left(2N\right)}^{b}+M\;\sigma_{m}.$$

In total, six ultrasonic fatigue tests at 20 kHz have been done with non-zero mean stress to estimate the parameters for Walker (Equation 2) and FKM (Equation 3) methods. This is shown in the Haigh diagram in Figure 11a in coordinates of mean stress $\sigma_{m}$ and stress amplitude $\sigma_{a}$. Four data points were obtained with 1.5 kN applied to specimens as a recommended maximum tensile force value; the remaining two data points had 2.0 kN and 2.6 kN applied, exceeding the recommended value. This set of tests resulted in the range of stress ratios R∈[−0.166,0.168] and a number of cycles $N\in [2.0\times 10^{4},7.2\times 10^{8}]$, as shown Figure 11b. The estimation of parameters for correction methods is done by manually matching the non-zero mean stress data points to the previously obtained data at zero mean stress. In other words, the data points are “compressed” into a single SN curve for *N* dependent on $\sigma_{ac}$, as shown in Figure 11b. This procedure resulted in the values of γ = 0.51 for Walker (Equation 2) and *M* = 0.42 for FKM (Equation 3) methods. With these values, it becomes possible to make predictions of fatigue behaviour for different stress ratios *R* in the domain where tension dominates. For example, Figure 11b illustrates the predicted SN curves for *R* = 0 using the Walker and the FKM methods. The only difference between these methods is a small discrepancy in the predicted fatigue limits of 242.3 MPa and 239.6 MPa, respectively. As seen in Figure 11a, the predictions of fatigue limit $\sigma_{lim}$ by both methods are very close in the range R∈[−1,0], but start to diverge very rapidly in the R>0 domain.

## 5. Conclusions

In this paper, the results of ultrasonic fatigue testing of structural steel S275JR+AR are reported. Steel S275JR+AR is a versatile material which can be used in a wide range of construction and engineering applications. However, it is most common for large components of heavy machinery used in the sand and aggregate and the mining industry, where it is expected to work for many years under continuous cyclic loading at relatively low-stress levels, but in harsh operating environments. This is the first research of its kind that is focused on the comprehensive fatigue investigation of the steel grade S275JR+AR in the gigacycle domain with number of cycles up to >$10^{10}$.

The insights into frequency, environmental and mean stress effects of the fatigue performance are also provided. In order to understand the influence of environment, the fatigue samples were tested in two surface conditions—polished and pre-corroded—for 2 weeks and 1 month. In order to reach a few billion cycles within a practically sustainable testing time, an accelerated fatigue testing is required. The goal of reaching gigacycle fatigue domain is achieved using the ultrasonic fatigue testing approach with the Shimadzu USF-2000A system, which runs at 20 kHz of resonance frequency.Confidence and competence in running ultrasonic fatigue tests have been obtained by overcoming multiple technical challenges. The duration of fatigue tests has been extended to reach 10 billion cycles in less than 10 days by modifying the forced cooling using a standard cooling hose with an additional cold-air gun able to reduce the temperature by 45 °C. However, this modification has introduced the instability of the cold air flow rate with unpredictable temperature fluctuation. The developed LabView fixed this issue. The program reads the temperature from the infrared PyroCube thermometer and initiates an emergency test stopping when temperature increases over 30 °C.Pronounced strain rate effect is a big challenge for the determination of SN curves with accelerated fatigue testing: particularly, the fatigue limit that strongly depends on the frequency of testing. Testing at 20 kHz using ultrasonic machines significantly exaggerates fatigue strength compared to normal loading conditions. For S275JR+AR steel grade, the quantitative difference between SN curves obtained at 15 Hz and 20 kHz was measured in terms of stress amplitude as 167.7 MPa on average.Basic quantification of the frequency effect contribution has been done using the estimated difference in stress amplitude between the high-frequency and low-frequency SN curves. A simple extrapolation/down-scaling of the ultrasonic fatigue testing results in a low-frequency domain that can be applied to the obtained data from pre-corroded samples. Consideration of the continuously progressing corrosion damage using pre-corroded batches of samples brings the extrapolated design stresses at 10 billion cycles to low and alarming values, which are still subject to validation.The fracture surfaces are investigated using both using optical stereo-microscopy and SEM microscopy. The advantage of the first method is that it can capture surface colouring that helps to understand the history of the crack propagation. The disadvantages are uneven focus and variable resolution quality for the whole fracture surface images. On the other hand, second method produces a perfect resolution images of the whole fracture surface, but colour information is completely lost.Finally, this research work has considered two mean stress correction methods—Walker and FKM. Both demonstrated a very similar effect on the available fatigue data extrapolation capability.

The obtained experimental results provide a solid background to understand the benefits and drawbacks of using steel with a relatively low yield strength and good ductility in the gigacycle fatigue domain and corrosive environments. The next step of this research work is to investigate the manufacturing effect by testing the samples cut from the welded plates with a heat-affected zone in the middle.

## Figures and Tables

**Figure 1 materials-16-01799-f001:**
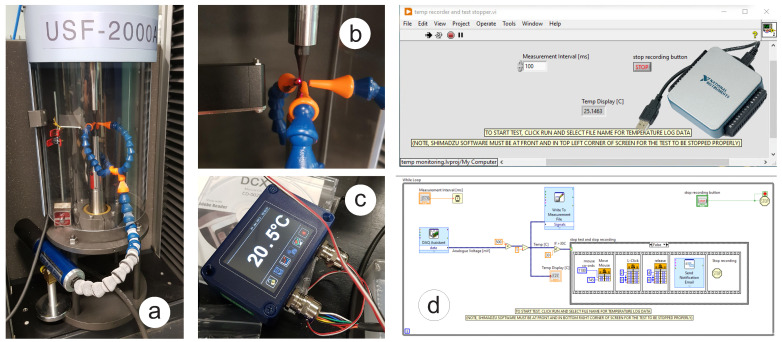
Modified arrangement for temperature control of the ultrasonic test: (**a**) USF-2000A with mean stress loading mechanism and VORTEC cold air gun; (**b**) Pyrocube IR temperature sensor; (**c**) touch screen display for PyroCube; (**d**) NI USB Multifunction DAQ data acquisition card and LabView program.

**Figure 2 materials-16-01799-f002:**
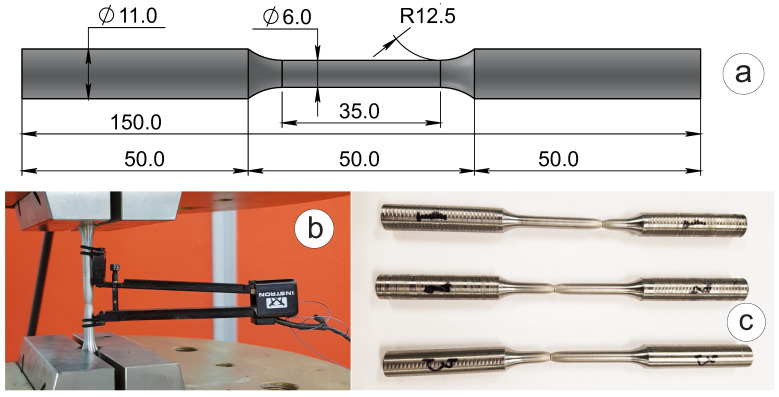
Tensile testing of S275JR+AR steel specimens with corresponding results in Table 1: (**a**) dimensions of the specimen; (**b**) specimen with the strain gauge attached; (**c**) necking of the specimens at failure.

**Figure 3 materials-16-01799-f003:**
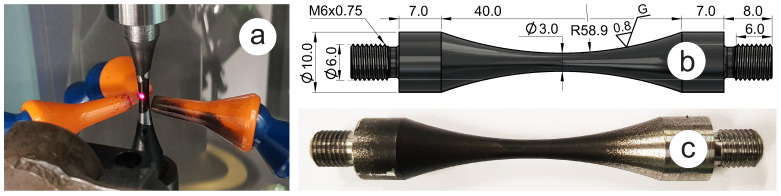
Specimen for ultrasonic test: (**a**) inserted in UFS-2000A with temperature measurement spot; (**b**) solid model with dimensions; (**c**) machined and painted specimen.

**Figure 4 materials-16-01799-f004:**
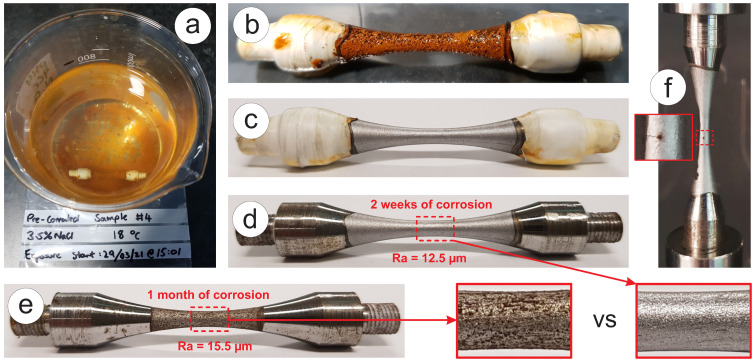
Pre-corroded specimen for ultrasonic test: (**a**) 3.5% NaCl solution with the submerged specimen; (**b**) rust layer on the specimen; (**c**) washed specimen with thread seal tape; (**d**) specimen after 2 weeks of corrosion; (**e**) specimen after 1 month of corrosion; (**f**) specimen at the end of the test with a crack in the middle.

**Figure 5 materials-16-01799-f005:**
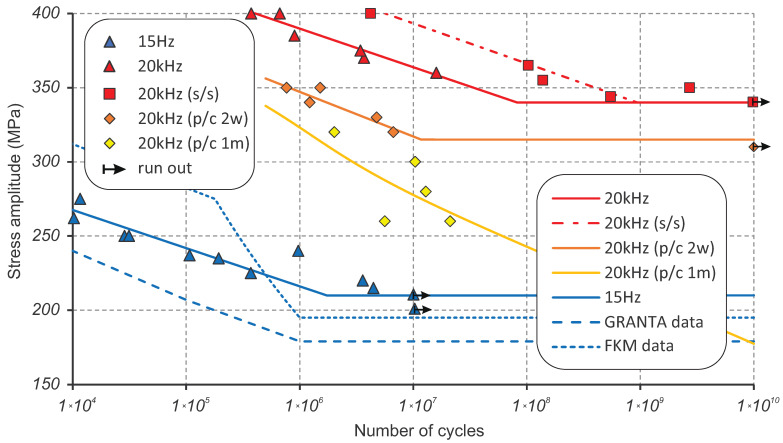
Summary of fatigue testing for S275JR+AR at 15 Hz and 20 kHz including base dry and pre-corroded samples.

**Figure 6 materials-16-01799-f006:**
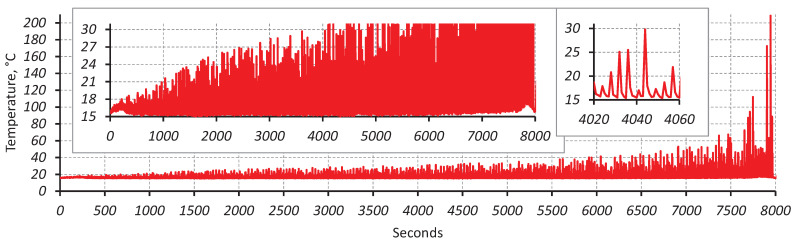
Temperature history of the ultrasonic fatigue test that lasted 4 million cycles at 400 MPa stress amplitude.

**Figure 7 materials-16-01799-f007:**
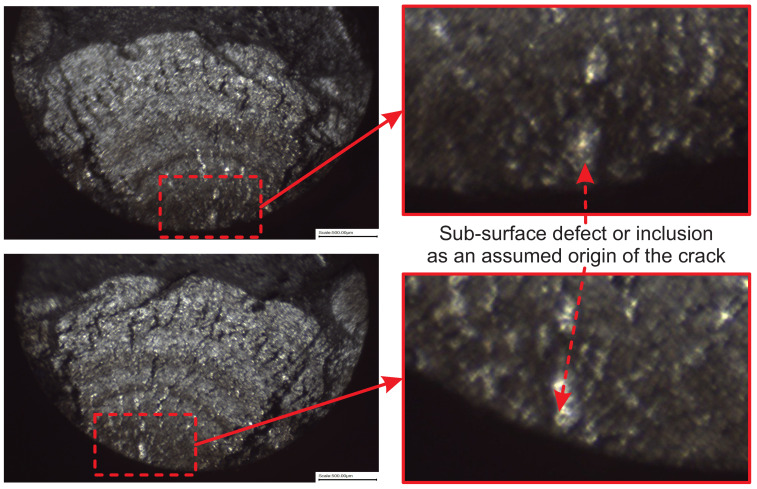
Search for the crack origin on the fracture surface of the sample run at 355 MPa stress amplitude for over 139 million cycles using Yenway optical stereo-microscope.

**Figure 8 materials-16-01799-f008:**
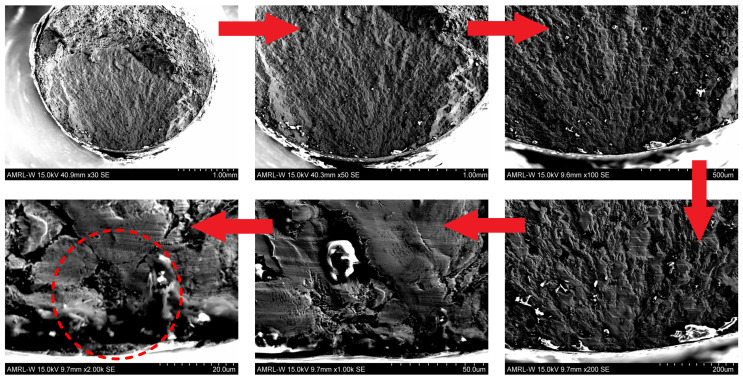
Search for the crack origin on the fracture surface of the sample run at 355 MPa stress amplitude for over 139 million cycles using Hitachi scanning electron microscope.

**Figure 9 materials-16-01799-f009:**
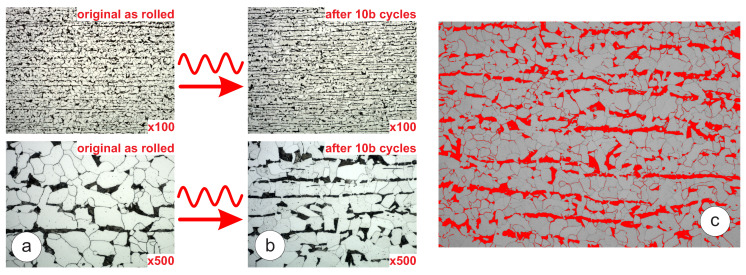
Microstructure of S275JR+AR steel extracted from the sample at the fatigue limit $\sigma_{lim}$ = 340 MPa: (**a**) before testing; (**b**) after 10 billion cycles; (**c**) processed micrograph showing the pearlite (red) and ferrite (white) regions.

**Figure 10 materials-16-01799-f010:**
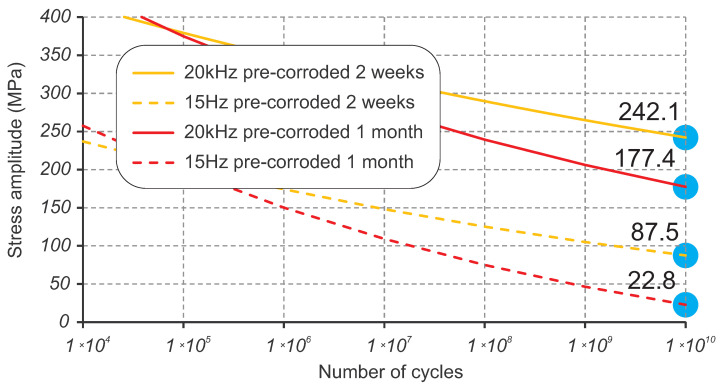
Extrapolation of the pre-corroded SN curves to 10 billion cycles and scaling down to low frequency.

**Figure 11 materials-16-01799-f011:**
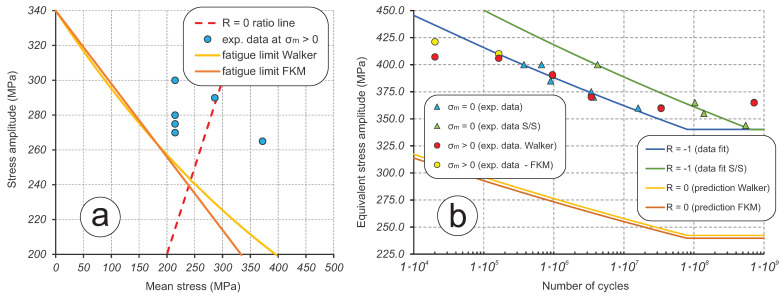
Investigation of the mean stress effect with Walker and FKM approaches: (**a**) Haigh diagram showing experimental data and fatigue limit extrapolation; (**b**) combined fatigue data for R=−1 and predicted SN curves for R=0.

**Table 1 materials-16-01799-t001:** Mechanical properties of 12 mm thick hot-rolled plate made of S275JR+AR structural steel.

Young’s Modulus [GPa]	0.2% Proof Stress [MPa]	Tensile Strength [MPa]	Elongation at Break [%]
211.1 (210 ‡)	314 (338 */275 †)	468.9 (469 */410 †)	31.8 (30.5 */23 †)

‡ according to the standard EN 1993-1-1 [20]. † according to the standard EN 10025-2 [1]. * from the material quality certificate provided by the manufacturer.

**Table 2 materials-16-01799-t002:** Material parameters of the SN curves for the least squares regression lines shown in Figure 5.

Groups	0	1	2	3	4
Parameters	15 Hz eq.fit	20 kHz eq.fit	20 kHz (s/s) eq.fit	20 kHz (p/c 2w) eq.fit	20 kHz (p/c 1m) eq.fit
*b*	−0.047	−0.03	−0.031	−0.039	−0.065
$\sigma_{f}^{\prime }$	426.15	599.71	659.12	610.56	829.06
$R^{2}$	0.92	0.93	0.98	0.84	0.44

## Data Availability

The data presented in this study are available on request from the corresponding author. The data are not publicly available due to business restrictions.

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
