# Peer review of "Ultrasonic Fatigue Testing of Structural Steel S275JR+AR with Insights into Corrosion, Mean Stress and Frequency Effects"

_materials, 2023, doi:10.3390/ma16051799_

Round 1
Reviewer 1 Report
The article reports on investigation of frequency, corrosion and mean stress effects on fatigue of S275JR+AR structural steel. The text is well organized and overall clearly written. The results obtained are presented in an appropriate form (with some minor exceptions discussed below) and, in the opinion of the reviewer, are beneficial to the scientific and engineering community. The article deserves publication, considering the recommendations below:
1. Figure 1 caption: should be ‘…mean stress loading mechanism…’
2. In the section ‘4.3 Strain-rate effect’ it is stated that: ‘The ferrite content of the materials was then compared to the change in the fatigue limit at 20Hz and 20kHz in order to evaluate the effect of the ferrite content on the strain rate sensitivity. It can therefore be seen that the material containing the higher ferrite content (S275JR) had the greater discrepancy between the SN curves at the two test frequencies and therefore had a higher strain-rate sensitivity.’ However, there is no visual or numerical proof of this statement provided in the paper, from which one could deduce higher susceptibility of S275JR+AR to frequency effect than other materials could have. Please provide such data in your paper.
3. It is unclear which frequency is used for non-zero mean stress fatigue tests.
4. Several times ‘S275JR+AS’ is used instead of ‘S275JR+AR’. Please correct that in the paper.
5. The text requires minor language correction.
Author Response
Dear Reviewer,
Thank you for your time and lots of efforts spent on careful reading, reviewing and constructive commenting on our manuscript ID no. materials-2121205.
As advised by the Editor, the manuscript went through the extensive English proofreading by all the co-authors. The text of the manuscript has been significantly improved stylistically and grammatically including both Abstract and Conclusions. The most significant changes are highlighted in blue colour. Five new references have been added to the manuscript to facilitate the discussion of
- size-effect in fatigue testing [Ref.21],
- crack initiation location [Ref.23 & 24],
- effect of the ferrite content on frequency sensitivity [Ref.29],
- extrapolation of the corrosion-fatigue results [Ref.30].
More specific comments and remarks are addressed below.
Reviewer:
- Figure 1 caption: should be ‘…mean stress loading mechanism…’
- In the section ‘4.3 Strain-rate effect’ it is stated that: ‘The ferrite content of the materials was then compared to the change in the fatigue limit at 20Hz and 20kHz in order to evaluate the effect of the ferrite content on the strain rate sensitivity. It can therefore be seen that the material containing the higher ferrite content (S275JR) had the greater discrepancy between the SN curves at the two test frequencies and therefore had a higher strain-rate sensitivity.’ However, there is no visual or numerical proof of this statement provided in the paper, from which one could deduce higher susceptibility of S275JR+AR to frequency effect than other materials could have. Please provide such data in your paper.
- It is unclear which frequency is used for non-zero mean stress fatigue tests.
- Several times ‘S275JR+AS’ is used instead of ‘S275JR+AR’. Please correct that in the paper.
- The text requires minor language correction.
Authors:
- Corrected
- Section “4.3 Strain-rate effect” has been modified accordingly to your request. An average ferrite content for S275JR+AR has been found as 82%. More discussion on the effect of the ferrite content on the frequency sensitivity and the heat generation is provided in our new conference publication:
https://doi.org/10.1016/j.prostr.2022.12.079
- Ultrasonic frequency – 20 kHz. Added to the last paragraph of Section 4.4.
- Corrected
- Corrected
Sincerely yours,
Yevgen Gorash on behalf of the authors
Reviewer 2 Report
Overall a very well prepared and interesting manuscript that is of great interest to the specific academic field, despite this there remains a few additions that could be made to improve the overall quality and compreshensiveness of the manuscript.
Figure 5: What is the source of the discrepancy between the FKM/Granta data and the test data presented? Is the strain rate effect (i.e. higher test frequency) really that beneficial to the fatigue life.
Fig 6, instead of presenting the temperature history as a function of time, which shows alot of higher frequency oscillations in the response, potentially consider doing a standard deviation of temperature vs time plot which highlights the increase in peak to valley temp towards the end of the test.
Figure 7, the yenway optical microscope images are quite poor in quality and the contrast/lighting is not ideal. Especially at the higher magnificiation the details are difficult to see, and the scale of the sub surface defect is almost macroscopic, so it might be difficult to definitively say what the source of the Fatigue Crack initiation is. The SEM images are much better and improve the overall understanding. The authors are encouraged to consider including (10.1016/j.ijfatigue.2018.06.028, and 10.1016/j.ijfatigue.2017.06.038) which have similar fractographic features at the fatigue crack initiation sites which helps to confirm the findings of the proposed manuscript.
Author Response
Dear Reviewer,
Thank you for your time and lots of efforts spent on careful reading, reviewing and constructive commenting on our manuscript ID no. materials-2121205.
As advised by the Editor, the manuscript went through the extensive English proofreading by all the co-authors. The text of the manuscript has been significantly improved stylistically and grammatically including both Abstract and Conclusions. The most significant changes are highlighted in blue colour. Five new references have been added to the manuscript to facilitate the discussion of
- size-effect in fatigue testing [Ref.21],
- crack initiation location [Ref.23 & 24],
- effect of the ferrite content on frequency sensitivity [Ref.29],
- extrapolation of the corrosion-fatigue results [Ref.30].
More specific comments and remarks are addressed below.
Reviewer:
- Figure 5: What is the source of the discrepancy between the FKM/Granta data and the test data presented? Is the strain rate effect (i.e. higher test frequency) really that beneficial to the fatigue life.
- Fig 6, instead of presenting the temperature history as a function of time, which shows a lot of higher frequency oscillations in the response, potentially consider doing a standard deviation of temperature vs time plot which highlights the increase in peak to valley temp towards the end of the test.
- Figure 7, the yenway optical microscope images are quite poor in quality and the contrast/lighting is not ideal. Especially at the higher magnificiation the details are difficult to see, and the scale of the sub surface defect is almost macroscopic, so it might be difficult to definitively say what the source of the Fatigue Crack initiation is. The SEM images are much better and improve the overall understanding. The authors are encouraged to consider including (10.1016/j.ijfatigue.2018.06.028, and 10.1016/j.ijfatigue.2017.06.038) which have similar fractographic features at the fatigue crack initiation sites which helps to confirm the findings of the proposed manuscript.
Authors:
- FKM and Granta do not provide an original experimental data. Other assumption is that these databases provide a “worst-case scenario” of the SN curves (design SN curves) with averaging the fatigue data over different subgrades / casts / batches. Yes, indeed higher testing frequency makes the materials to look stronger. This has been recently confirmed by us through the high-strain-rate tensile testing, there the yield strengths values were much higher compared to standard values. Writing of the corresponding manuscript is now in progress in the team.
- This is an excellent idea, thank you very much! We have very similar ideas in regards how to investigate deeper into the temperature history. The most important is that we have recently become more skilful in obtaining higher quality temperature data by using National Instrument acquisition hardware in combination with LabView application. This has let us to record the temperature history with the resolution of 0.001s. However, it has created the next challenge of processing the data, as its volume has increased massively. Fore this purpose, we have started using Python platform, which is much better suited to work with “big data”. We are not using Excel any more for the temperature data processing for our current work. Again, the progress in this area of research will be provided in the future publication.
- Thank you very much for your comment. The suggested references have been added to the manuscript.
Sincerely yours,
Yevgen Gorash on behalf of the authors
Reviewer 3 Report
This manuscript needs to be revised as most of the content is already published in Procedia in 2022. The new results on corrosion effect and mean stress effect to be high lighted.
Conclusions need to be revised. the first few lines (350-356) may be removed as it reflects introduction.
Author Response
Dear Reviewer,
Thank you for your time and lots of efforts spent on careful reading, reviewing and constructive commenting on our manuscript ID no. materials-2121205.
As advised by the Editor, the manuscript went through the extensive English proofreading by all the co-authors. The text of the manuscript has been significantly improved stylistically and grammatically including both Abstract and Conclusions. The most significant changes are highlighted in blue colour. Five new references have been added to the manuscript to facilitate the discussion of
- size-effect in fatigue testing [Ref.21],
- crack initiation location [Ref.23 & 24],
- effect of the ferrite content on frequency sensitivity [Ref.29],
- extrapolation of the corrosion-fatigue results [Ref.30].
More specific comments and remarks are addressed below.
Reviewer:
- This manuscript needs to be revised as most of the content is already published in Procedia in 2022. The new results on corrosion effect and mean stress effect to be highlighted.
- Conclusions need to be revised. the first few lines (350-356) may be removed as it reflects introduction.
Authors:
- Current revision of the manuscript contains more details on the research and more than 50% of new text content compared to the conference paper in Proceedia Structural Integrity 2022.
- Conclusions have been modified.
Sincerely yours,
Yevgen Gorash on behalf of the authors
Reviewer 4 Report
This is an excellent job. The idea of utilizing fatigue in the ultrasonic range is highly innovative. All the phenomena are properly explained with text and photographs. All the scientific arguments are substantiated by mathematical equations.
The treatise is worth reading.
Author Response
Dear Reviewer,
Thank you for your time and lots of efforts spent on careful reading, reviewing and constructive commenting on our manuscript ID no. materials-2121205.
As advised by the Editor, the manuscript went through the extensive English proofreading by all the co-authors. The text of the manuscript has been significantly improved stylistically and grammatically including both Abstract and Conclusions. The most significant changes are highlighted in blue colour. Five new references have been added to the manuscript to facilitate the discussion of
- size-effect in fatigue testing [Ref.21],
- crack initiation location [Ref.23 & 24],
- effect of the ferrite content on frequency sensitivity [Ref.29],
- extrapolation of the corrosion-fatigue results [Ref.30].
More specific comments and remarks are addressed below.
Reviewer:
This is an excellent job. The idea of utilizing fatigue in the ultrasonic range is highly innovative. All the phenomena are properly explained with text and photographs. All the scientific arguments are substantiated by mathematical equations. The treatise is worth reading.
Authors:
Thank you very much for your positive feedback! We plan to publish more on recently accomplished research related to ultrasonic fatigue testing of other structural steels and their welds.
Sincerely yours,
Yevgen Gorash on behalf of the authors
Reviewer 5 Report
COMMENTS TO THE AUTHOR(S)
The submitted research on “Ultrasonic fatigue testing of structural steel S275JR+AR with insights into corrosion, mean stress and frequency effects” has presented an interesting elaborative work. I congratulate the author(s) for the same and wish them best of luck for their future endeavours.
1. Conclusions should be strictly point-wise and need to increased in number i.e., atleast from each results and discussion sub-section/paragraph, a specific conclusion needs to be drawn individually.
2. The discussed correlation on effect of corrosion to ultrasonic fatigue needs more basis such as microstructural manifestations in order to interpret the same scientifically.
3. Related to comment 2, author(s) should take support from literature to support their arguments in that regard.
*Author(s) should highlight all the modifications carried out in the paper.
Author Response
Dear Reviewer,
Thank you for your time and lots of efforts spent on careful reading, reviewing and constructive commenting on our manuscript ID no. materials-2121205.
As advised by the Editor, the manuscript went through the extensive English proofreading by all the co-authors. The text of the manuscript has been significantly improved stylistically and grammatically including both Abstract and Conclusions. The most significant changes are highlighted in blue colour. Five new references have been added to the manuscript to facilitate the discussion of
- size-effect in fatigue testing [Ref.21],
- crack initiation location [Ref.23 & 24],
- effect of the ferrite content on frequency sensitivity [Ref.29],
- extrapolation of the corrosion-fatigue results [Ref.30].
More specific comments and remarks are addressed below.
Reviewer:
- Conclusions should be strictly point-wise and need to increased in number i.e., at least from each results and discussion sub-section/paragraph, a specific conclusion needs to be drawn individually.
- The discussed correlation on effect of corrosion to ultrasonic fatigue needs more basis such as microstructural manifestations in order to interpret the same scientifically.
- Related to comment 2, author(s) should take support from literature to support their arguments in that regard.
Authors:
- Conclusions have been modified.
- Our approach to the interpretation and implementation of the corrosion effect on the fatigue testing results is based on our previous research work (https://doi.org/10.1016/j.proeng.2018.02.063), which was focussed on the comprehensive fatigue testing of the steel S355J0 with a particular interest to the corrosion. The idea of extrapolation of corrosion-fatigue results into gigacycle domain has been developed that time. Our assumptions agree with the experimental results from previous work.
- We have added 5 new references to get more support from literature.
Sincerely yours,
Yevgen Gorash on behalf of the authors
Round 2
Reviewer 3 Report
The revised manuscript addresses issues raised in review report.